# Bioactive Components of Human Milk and Their Impact on Child’s Health and Development, Literature Review

**DOI:** 10.3390/nu16101487

**Published:** 2024-05-14

**Authors:** Hubert Szyller, Katarzyna Antosz, Joanna Batko, Agata Mytych, Marta Dziedziak, Martyna Wrześniewska, Joanna Braksator, Tomasz Pytrus

**Affiliations:** 1Student Scientific Group of Pediatric Gastroenterology and Nutrition, Wroclaw Medical University, 50-369 Wroclaw, Poland; katarzyna.antosz@student.umw.edu.pl (K.A.); joanna.batko@student.umw.edu.pl (J.B.); agata.mytych@student.umw.edu.pl (A.M.); marta.dziedziak@student.umw.edu.pl (M.D.); martyna.wrzesniewska@student.umw.edu.pl (M.W.); 22nd Clinical Department of Paediatrics, Gastroenterology and Nutrition, Wroclaw Medical University, 50-369 Wrocalw, Poland; joanna.braksator@umw.edu.pl (J.B.); tomasz.pytrus@umw.edu.pl (T.P.)

**Keywords:** breast milk, breastfeeding, immune system, infant, bioactivity, development

## Abstract

The composition of human breast milk is an ideal combination of substances necessary for the healthy development of an infant’s body while protecting from pathogens and the balanced development of the microbiota. Its composition is dynamic and changes with the age of the child, meeting their current needs. The study provides a thorough overview of human milk components, such as immunological components, growth factors, hormones, carbohydrates, lipids, minerals, and vitamins. Authors focus on capturing the most important aspects of the effects of these substances on a newborn’s body, while also looking for specific connections and describing the effects on given systems. Supplementation and the use of ingredients are also discussed. The purpose of this paper is to present the current state of knowledge about the bioactive components of human milk and their impact on the growth, development, and health of the young child.

## 1. Introduction

According to the WHO, feeding exclusively on human milk (HM, BM) is widely recognized as the gold standard of nutrition for a child up to 6 months of age, and in the second half of the first year of life, it can provide more than half of the required nutrients [1]. Breastfeeding can continue even up to the second year of life, assuming it supplements the infant’s diet with additional enriched substances that are in short supply in milk, such as iron [1,2,3]. In addition, breastfeeding has numerous advantages unrelated to its composition: it is cheap, readily available, and uncontaminated, and the very act of breastfeeding is an important experience in building the bond between a mother and child [4]. The breastfeeding scheme recommended by the WHO is shown in Figure 1.

The composition of human milk is not constant and changes, adapting to the diet of the mother and the age of the fed child, but roughly speaking it can be considered to consist of 87% water, 3.8% fats, which are the main energy resource of milk, 1.0% protein, and 7% lactose [4].

The most important function of human milk is to enable the child’s body to pass through the postnatal period, through the time of the body’s greatest vulnerability to pathogenic microorganisms and deficiencies resulting from a body that is just developing, until it is fully functional. For this reason, it contains numerous and diverse bioactive elements affecting the endocrine, immune, digestive, and respiratory systems. Learning about the bioactive components of human milk and their effects on the baby’s body is the undeniable core of the development of neonatal care [5,6]. Human milk feeding still remains a taboo subject in many circles [7,8]. In Europe, only 25% of the women breastfeed exclusively until the infant is 6 months old [9]. The situation is similar in the U.S., where although as many as 81% of the women begin breastfeeding after giving birth, only 25.5% breastfeed exclusively until 6 months [10].

In this research study, the electronic database searches were focused on breast milk, its compounds, as well as its impact on newborns. The authors looked at original articles, meta-analyses, clinical cases, and systematic reviews that are connected with the topic, which were found in the PubMed database. The keywords are described above. The literature search was conducted with the PubMed database and 134 works accessed before April 2024 were selected, mostly within 10 years. Moreover, additional significant studies are included too.

**Figure 1 nutrients-16-01487-f001:**
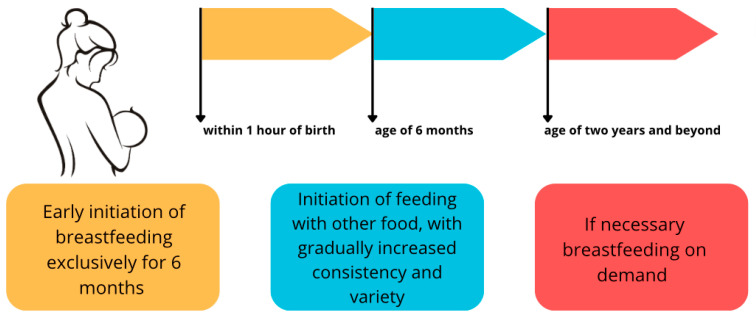
Breastfeeding scheme according to the WHO recommendations [1,2,3]. An original author’s artwork.

## 2. Compounds of Mainly Immunological Importance

The period immediately following birth is extremely dangerous for the newborn baby because, having an undeveloped and immature immune system, he or she is suddenly exposed to an enormous number of pathogenic microorganisms in his or her environment, possessing only the antibodies previously supplied by the placenta with the mother’s blood. For this reason, colostrum, the milk produced in the first days after birth, contains more bioactive immunological and anti-inflammatory components than transitional and mature milk, providing passive immunization from the first days of the baby’s life [11,12].

### 2.1. Antibodies

SIgA, a subtype of IgA antibodies, constitutes the most abundant component of the mode of immunologically active substances contained in human milk, which is their only source during the first four weeks of a newborn’s life [13]. They are produced by the B lymphocytes of the lymphatic system in the mammary gland due to the functioning of the so-called gut–mammary gland axis (MG-secretory IgA axis). Its function is based on contact with the pathogen in the lumen of the mother’s intestine and, via the M cells lying between the enterocytes, the transmission of the antigen to the dendritic cells contained in the Peyer’s patches, which belong to the gut-associated lymphoid tissue (GALT) [14,15]. The dendritic cells then present the antigen to T lymphocytes and B lymphocytes, inducing the second one to transform into plasma cells. The resulting plasmoblasts migrate to the lymph nodes of the mammary glands with the involvement of the chemokine CCL28, where they complete the transformation to plasmocytes and produce sIgA antibodies tailored to the pathogens to which the child’s mother has previously been in contact and to which the child is also likely to be exposed [13,15,16].

The content of sIgA varies from 5 mg/mL in colostrum to 1 mg/mL later in feeding, accounting for approximately 90% of all antibodies; the baby takes in 0.3 g/kg/day of which only 10% enter the bloodstream [11,17]. This is because they perform their function locally, on the surface of the mucous membranes, in a mechanism called immune exclusion [13] by binding commensal microorganisms and pathogens such as bacteria, viruses, and bacterial lipopolysaccharides. In this way, they prevent their adhesion and penetration into the organ epithelium and consequently the formation of infectious inflammatory reactions. Their protective function is mainly focused on the gastrointestinal and respiratory systems, thus constituting the body’s first line of defense against pathogens causing infectious diarrhea, inflammatory bowel disease, necrotizing enterocolitis, and ARI [17,18,19,20], which are diseases that contribute to high neonatal mortality. To a lesser extent, antibodies in the IgM class have a similar protective function for the mucosa [11,17,20].

Human milk also contains much less abundant antibodies in the IgG class, whose function is not yet well understood. Unlike IgA and IgM, they do not protect the mucous membranes and penetrate the epithelial barrier to a negligible extent. It has been suggested that IgG antibodies not only act as anti-inflammatory agents [13] but also, when co-operating with sIgA antibodies contained in milk, increase immune tolerance through the formation of IgG-allergen complexes, and thus present it to epithelial cells, reducing the risk of developing an allergy [17,20].

### 2.2. Lactoferrin, Lysozyme, Lactahedrin and Lactoperoxidase

Lactoferrin is a metal-binding glycoprotein present most abundantly in colostrum (5–7 g/L) and is also found in lower concentrations in mature milk (1–3 g/L) [19,21,22,23]. It exhibits bacteriostatic and immunomodulatory properties by eliminating factors that cause acute inflammatory responses and reducing the levels of the pro-inflammatory cytokines [13,23,24]. Isolated and purified lactoferrin in human milk has also been shown to have an inhibitory effect on the biofilm formation of the opportunistic bacterium *Pseudomonas aerugunosa*, which contributes, among other things, to bacteremia and septic shock among premature infants [25]. The high lactoferrin content of breast milk also has a positive effect on the clinical course of ARI [19].

Lysozyme, an enzyme that enables the degradation of the glycoproteins of the outer wall of Gram-positive bacteria, also exhibits antibacterial activity against Gram-negative bacteria, along with lactoferrin [11,13,26]. It has also been reported that both lysozyme and lactoferrin exhibit antiviral activity [27].

Milk fat globule-EGF factor 8 protein (Mfge8), also known as lactadherin, participates in immune processes mainly through a cascade that blocks NF-kB via TRL4 inhibition. It thus reduces the resulting inflammation and prevents pro-inflammatory reactions. Its function is mainly intestinal. It also shows an inhibitory effect on the development of symptoms in rotavirus infection, which is particularly dangerous for newborns [13,28].

An important function in the antimicrobial function of human milk is performed by alpha-lactalbumin, produced by the epithelial cells of the mammary glands [11,21,29,30]. This most-abundant protein of human milk (20–25% of all protein components) has several functions. It participates in the biosynthesis of lactose, enables the uptake of elements such as calcium and zinc into milk, and provides an adequate amount of amino acids essential for the newborn. It also forms HAMLET, Human Alpha-lactalbumin Made Lethal To Tumor Cell complex, which stimulates the apoptosis of cancer cells, providing important anti-tumor protection for the infant [21,29,30]. Alpha-lactalbumin also shows antimicrobial [11] and antiviral activity [31].

The lactoperoxidase system (LPS) includes the enzyme lactoperoxidase, thiocyanate (SCN-) present in breast milk, and H_2_O_2_ present via bacterial metabolism. Lactoperoxidase uses H_2_O_2_ in order to oxidize SCN- to hypothiocyanite (OSCN-), which has antibacterial properties. Overexpressed OSCN- is absorbed by the antioxidants in human breast milk, limiting local tissue damage by oxidative stress [32]. 

### 2.3. Cytokines

Cytokines are small protein molecules that are an important part of the body’s immune system; they are not only a marker of inflammation but, above all, synchronize the body’s immune response and regulate the development and inhibition of inflammation resulting from infection [29]. Both pro-inflammatory and anti-inflammatory cytokines are present in breast milk, originating from the epithelial cells of the mammary glands, produced by immune cells contained in the milk solution (leukocytes such as macrophages, neutrophils, and lymphocytes) or originating from the mother’s bloodstream [33,34,35].

Beyond their immunoregulatory function in every organism, some ILs play a special role in the context of human milk formation and human milk bioactivity. The selected cytokines are summarized in Table 1. 

The concentration of interleukins depends on a number of factors such as the stage of lactation, the health of the mother, and the health of the newborn. Colostrum is particularly rich in IL-6 and TGF-β; in the case of maternal allergy, the concentration of TGF-β decreases and the concentration of the cytokines IL-4 and IL-10 increases. Changes in the amount of cytokines are also influenced by maternal stress and diet [33,35].

## 3. Growth Factor Family

Another significant and comprehensive group of human milk (HM) compounds is the growth factor family. A growth factor (GF) is a natural bioactive compound that is capable of stimulating cell proliferation and differentiation. GF interaction with corresponding structures leads to an evolution in very different systems, e.g., the intestinal tract, vasculature, nervous, and endocrine ones. These structures can interact with receptors to induce various types of biochemical responses to promote cell development, which naturally seems to be an important issue when it comes to newborns. Moreover, these compounds are at their highest peak of concentration exactly in the early postnatal period, which also underlines their significant role in the first days of newborns [39].

In this section, the following will be discussed: EGF, HB-EGF, VEGF, EPO, IGF, NGF, and G-CSF. Additionally, TGF- β described above can be considered as a member of the GF family. The described GFs and their effects are described in Table 2.

### 3.1. Epidermal Growth Factors

Starting with EGF, whose role is mostly focused on the infant’s intestine mucosa. EGF plays a role in the maturation of intestine structures, it helps with the limitation of pathogenic bacterial growth, and it is capable of succoring the immune system against systematic infections [38,40].

As with many other compounds, EGF changes its concentration during breastfeeding time. The very meaningful period of the first days is connected with the highest EGF concentration in human colostrum. This concentration decreases slowly but steadily in HM during the first two months of lactation to reach half of the initial value [40,41]. What is worth highlighting, the concentration of EGF is relevantly higher in the HM of premature babies’ mothers. This difference is connected with the biological response for the insufficient prenatal development of baby intestines, as there was not enough time for EGF contained in amniotic fluid to perform [42].

The structure of EGF is durable to low pH and a majority of digestive enzymes. This property allows EGF to pass through the digestive system until the intestine. After finding the corresponding receptors (EGF receptors—EGFRs), the enterocytes are stimulated to increase their production of proteins and DNA, to divide and develop themselves, and to absorb water and glucose [39,43].

Additionally, to its protective function on the intestine, EGF promotes the proper function of the intestinal barrier. EGF protects the quality of the tight junction between cells, as well as the actin cytoskeleton. Furthermore, it reduces the permeability of the epithelial barrier, which can be caused by hydrogen peroxide (due to mitogen-activated protein kinase (MAPK)-dependent mechanisms) [44].

While EGF is responsible for the well-being of intestine mucosa, it is considered one of the protective factors against necrotizing enterocolitis (NEC). NEC is one of the most common diseases when it comes to premature infant intestine conditions. Due to its high mortality rate (20–30%), it is considered a serious threat to newborn development. Therefore, the exact and physiological concentration of EGF, its activity and function, as well as the higher concentration in cases of strongly premature babies (23–27 weeks of gestation) is essential for proper intestine function [41,42,45].

As it is connected with the protection and regeneration of the epithelial layer, as well as with its stability in the gastric system, EGF is considered to be supplemented for the most high-risk newborns [41].

### 3.2. Heparin-Binding EGF-like Growth Factor (HB-EGF)

Heparin-binding growth factor (HB-EGF) is a member of the EGFR family of ligands [39,41]. HB-EGF is secreted by, e.g., macrophages (firstly described), fibroblast, smooth muscle, and epithelial cells, and can be found both in HM and amniotic fluids, as well as EGF. This versatility in the localization of production and function emphasizes the significant role of GFs during the earliest steps of human development [41].

The concentration of HB-EGF is significantly lower in comparison to EGF. However, their binding capacity to EGFR is incomparably higher. Another difference in this case is their permanent concentration during lactation [46].

Along with EGF, HB-EGF stimulates the proliferation, development, and regeneration of epithelial cells. It also helps fortify the intestine barrier. While binding the EGFR, HB-EGF mediates cell migration. Moreover, both HB-EGF and EGF reduce local inflammation and cell apoptosis [41].

In the cases of perfusion disorders, the concentration of HB-EGF immediately increases to support epithelial cell renewal. This upregulation is due to biological pathways such as Phosphoinositide 3-kinase (PI3K/Akt) and mitogen-activated protein kinase/Extracellular signal-regulated kinase (MEK/ERK1/2) [47]. This compound seems to indicate an active role in intestinal vessels and microvessels, which undoubtedly is vital for newborns and could have an application in the future [41].

### 3.3. Vascular Endothelial Growth Factor (VEGF)

While creating the human body from scratch, angiogenesis is one of the basic and necessary elements. The formation of new vessels is regulated via the expression and function of VEGF. The concentration of VEGF, as well as in the case of EGF, is at the highest point in colostrum. What is quite interesting, the concentration of VEGF is almost twice as high in term HM than in the preterm one. Additionally, with the time of lactation, this value decreases [47,48].

VEGF is a member of a large family of factors such as VEGF-A, B, C, D, E, and PIGF (placental-derived growth factor). VEGF-A is commonly labeled as VEGF and is the most valid in terms of vascular angiogenesis [49].

The function of VEGF is based on the interaction with the right receptor (VEGFR2) and the state of hypoxia. In a low-oxygen environment, with the participation of hypoxia-inducible factor (HIF), there is an increased production of VEGF. Sequentially, binding the mentioned receptor, VEGF contributes to endothelial cell development [50].

Last but not least, the concentration changes in VEGF are considered a significant variable in the pathogenesis of the retinopathy of prematurity (ROP). In this state, due to the pulmonary immaturity that is conjugated with oxygen therapy, and therefore the negative VEGF regulation, there are ideas to supplement VEGF via HM to help limit the developing ROP [51].

### 3.4. Erythropoietin (EPO)

The following factor is also related to the condition of hypoxia. Erythropoietin (EPO) stimulates erythropoiesis, especially in a state of cellular hypoxia.

EPO is a hormonally active glycoprotein. Its production takes place in the liver and renal cortex, and then, via the corresponding receptor (EpoR), it helps with raising the level of red blood cells (RBCs).

EPO can also be found both in HM and amniotic fluid, where its levels fluctuate according to hypoxia states. After birth, in HM the level of EPO is evaluated during lactation, increasing with the time of breastfeeding. Additionally, it seems that the time of delivery and duration of pregnancy have no meaningful connection with EPO concentration [39,41,52].

The idea of using EPO in trials is based on its properties. Pathological states, such as blood loss or digestive system diseases, may lead to lower RBC levels. There are suggestions that the supplementation of EPO could help newborns improve their conditions more efficiently and faster. However, test results have stayed unclear, and definitely, more data are needed [53,54].

### 3.5. Insulin-like Growth Factor (IGF)

Another large group of compounds to be found in HM is the insulin-like growth factor (IGF) superfamily. In this group, IGF-I, IGF-II, or IGF-binding proteins (IGFBPs) and IGF-specific proteases can be distinguished [38].

In human serum, IGF-I and IGF-2 are mostly bonded with IGFBP, which improves these hormones’ stability and activity. The levels of these agents are the highest in the colostrum, and then they start to decrease with the time of lactation [55]. Additionally, there are no noticeable differences in a subject of preterm and term HM, with only one big change of IGFBP-2 being in higher concentration in preterm milk, which also can be due to the mentioned better bioactive parameters of IGF-I, IGF-2, and IGFBP complexes [56].

The studies show the connection between breastfeeding and IGF-I concentration in infants, both in the digestive tract and blood serum. Moreover, this higher concentration in the cases of breastfeeding can be linked with the improved regulation of growth stimulation and guide differentiation in the first years of life. These regulations define infant growth in terms of fat accumulation, and therefore BMI in the first years [57].

Lastly, trials for enteral IGF-I applications are taken, e.g., to stimulate erythropoiesis, but their results remain unclear. More data are required [58].

### 3.6. Granulocyte Colony-Stimulating Growth Factor (G-CSF)

Granulocyte Colony-stimulating growth factor (G-CSF) is present not only in HM but also from the very first stage of fetus formation, where it is produced by the fetus itself and the placenta. G-CSF is responsible for proper inflammation response, as well as for the maintenance of hematopoiesis, and of course, for leukopoiesis. The trials of G-CSF are quite thoroughly described, and plenty of data are available. Considering its function, it is easy to predict that higher levels of G-CSF can be detected in amniotic fluid and the colostrum, as the proper leukocyte values are necessary from the first days of infant life [41].

The function of orally taken G-CSF is strongly connected with the infant’s intestines, and via specialized receptors, it can influence the growth, integrity, and differentiation of the digestive tract [41].

Taking into consideration its function, G-CSF could be applied in many therapies to avoid or treat infant states such as neutropenia or sepsis. While speaking of applications, it is worth highlighting that the resistance to the gastric environment of both human and recombinant G-CSF is satisfactory, while the agents added to infant formula seem to have no proper protection [57].

Although in the theory G-CSF application should protect infants, in real-life studies the data are confusing. There are results from a large cohort of newborns with neutropenia, in which the administration of G-CSF seems to be connected with secondary sepsis in the state of the improvement of neutropenia [59]. Currently, G-CSF administration is not a common clinical practice, but it is something to be considered and in need of further data [41].

### 3.7. Hepatocyte Growth Factor (HGF)

Hepatocyte growth factor (HGF) is, according to its name, responsible for proper hepar formation, but not limited to that only. HGF can promote organogenesis in other systems by stimulating the development of lungs, muscles, kidneys, mammary glands, and neurons. HGF acts by paracrine and endocrine signalization [58,60].

In HM a high concentration of HGF can be detected, which can be easily connected with the importance of the evolution of an infant’s organs [58]. In addition to its function of stimulating cell proliferation, HGF also regulates VEGF synthesis. The dual interactions of growth factors are very common in terms of human formation, as this highlighted example [61].

### 3.8. Neurotrophic Factors (NFs)

Last, but not least, Brain-derived neurotrophic factor (BDNF) must be described. These tiny neurotrophic factors (NFs) are strongly widespread in the adult brain. BDNF, in the company of a similar agent, glial cell line-derived neurotrophic factor (GDNF), and S100 B protein, significantly impacts the nervous system’s final formation [58]. What is worth highlighting, the levels of S100 B protein and GDNF elevate with the lactation time [62].

BDNF and GDNF with another one from this family, ciliary neurotrophic factor (CNTF), are present in the detected concentration in HM even up to 3 months of lactation. Research shows that when consumed by infants, GDNF from HM increases nervous system formation through neuron survival and its growth stimulation [39,63].

Another aspect to consider in terms of NFs and breastfeeding is that proper nutrition for infants is connected with better nervous system development. There are studies of rodents that were growing with a lack of Nfs during their fetal life or their early stages of life after birth. These rodents present noticeable defects in their nervous system development, e.g., in the enteric nervous system, which occurs with the significant deterioration of peristalsis, or impaired memory [64,65].

**Table 2 nutrients-16-01487-t002:** Summary of the most important features of GFs.

Growth Factor	Role and Ways of Applications	Sources
EGF	stimulating the development of epithelium, mainly intestines and limiting the pathogenic bacterial growth, can be supplemented in NEC to protect the intestines of a premature baby	[38,40,41,42,45]
HB-EGF	stimulating the proliferation, development, and regeneration of epithelial cells, could potentially be used in the perfusion disorders	[41,47]
VEGF	a strong factor in angiogenesis, considered for the prevention of ROP	[47,49,51]
EPO	stimulating erytropeiesis, used in the cases of lower RBC levels	[52,53,54]
IGF	stimulating the proliferation of stem cells, promoting the survival of cells, reducing or decreasing apoptosis, and the attenuation of inflammations, its high concentration in breast milk has a positive effect not only on the development of the newborn but also on BMI	[38,55,56,57]
G-CSF	responsible for proper inflammation response, the maintenance of hematopoiesis, and leukopoiesis, applied in states such as neutropenia or sepsis	[66,67]
HGF	responsible for proper hepar formation, as many others, the role of breastfeeding in supporting the final formation of the organs	[58,61]
NFs	impact on the nervous system’s final formation, breastfeeding supports proper nervous and mental development	[58,62,64,65]

## 4. Hormones

Breast milk (BM) is not only rich in nutrients, but it also contains multiple hormones, the content of which varies depending on the time of day but is also different in the postpartum period and different after the return of menstruation. All those hormones are responsible for providing sufficient infant growth and development. In the next paragraphs of our work, we discuss the most important hormones contained in breast milk and their functions. Their effects on BM are also described in Table 3.

### 4.1. Sex Steroid Hormones

Estrogens in breast milk can come from maternal circulation and local production. They pass from the mother’s body to breast milk by passive diffusion as they are small and lipophilic. Their production can also occur in the mammary gland due to the aromatization of androgens [68]. Breast milk contains estrone (E1), estradiol (E2), and estriol (E3), which are the most common naturally occurring estrogens [69]. E1 has the lowest concentration in human milk, and E3 has the highest level. The level of those hormones changes over time and studies show that their concentration is higher in the colostrum and lower to undetectable in mature milk. The exception is E3, the concentration of which increases in mature milk [68]. A single study reported that E2 from BM is positively correlated with an infant developmental score measuring eye movement, face recognition, the production of and responses to sound, attention to features, the holding of objects, and neck and feet movements [70]. However, the effects of BM estrogens on infants’ development are still not clear and need studying.

Dietary estrogens, called phytoestrogens, which are estrogenic compounds found in plants such as soybeans, flaxseed, fruits, vegetables, and cereals, have a similar structure to E2 and thus they can interact with ERs [71]. As phytoestrogens are found in dietary products, different populations are likely to have varying concentrations of BM phytoestrogens and therefore different infant exposure [68]. Studies on animals have shown that exposure to dietary phytoestrogens affects the reproductive system and thymus and causes impaired immune function, but also has a beneficial impact on obesity as it reduces weight and adipose tissue [68,72]. BM phytoestrogens can have a similar effect on infants, but more data are needed to be sure of the significance of this impact.

Progesterone, a 21-carbon steroid hormone, is transferred to BM from the maternal circulation or comes from sequestration in adipose tissue [73]. Progesterone concentration declines rapidly with time after birth with the lowest concentrations measured in mature milk [68]. Its levels depend on its concentration in maternal serum and possibly also on the mother’s diet; however, this matter needs further studying. BM progesterone might be negatively correlated with infant development score at 1 and 6 months postpartum and with infant weight but an investigation of the physiologic significance of BM progesterone is needed [70].

Testosterone, a key hormone of the male reproductive system, is also present in BM, at least for the first 6 months of breastfeeding, and its concentrations are similar in preterm and term breast milk. Beyond its function as a sex hormone, testosterone also has a neuro- and immunomodulatory role, increasing hemoglobin and hematocrit levels; thus, it is an important ingredient of BM [74].

### 4.2. Thyroxine and Thyroid-Stimulating Hormone

Thyroid-stimulating hormone (TSH) is a hormone produced by hypophysis. It stimulates the thyroid to produce its hormones. It was found that TSH is measurable during the first 6 months of lactation; however, it is not clear if TSH is bioavailable if provided orally [75].

Thyroxine (T4) and 3,5,3′-triiodothyronine (T3), hormones produced by the thyroid gland, whose function is controlling the metabolism in various cells and tissues, especially in the central nervous system (CUS), are also present in BM. It was found that BM can provide a significant exogenous source of T4 and T3 to the infant and their concentration in BM rises in the first 8 weeks postpartum [76,77]. However, studies show that although the amount of thyroid hormones in BM may delay the clinical recognition of hypothyroidism in infants, it is insufficient to prevent its detrimental effects [76].

### 4.3. Glucocorticoids

Among many of the bioactive factors, breast milk contains glucocorticoids (GCs): cortisol, cortisone, and corticosterone GCs are transferred to BM through simple diffusion [78]. In studies, increased breast milk cortisol concentrations in the first 12 weeks of an infant’s life were found [79]. In BM, GCs show circadian variation with their highest concentration found in the morning [80].

GCs from BM are essential for infants. They help to regulate the infant’s cortisol levels and various physiological processes and promote healthy growth and development [79]. Studies on primates and humans showed that breast milk cortisol was associated with infant temperament and fear reactivity [81,82]. Higher concentrations of cortisol may have a negative effect on the immune system as higher levels of cortisol are associated with lower levels of sIgA [13]. However, some studies found that milk cortisol is positively associated with pro-inflammatory responses to some bacteria in vitro [83]. Animal studies also suggest that cortisol may improve intestinal development in both the microbiome and intestinal immune system [78].

Although there were many studies covering the role and regulation of GC concentration in BM, their results need proving and further studies have to be conducted in order to learn about the actual impact of the GCs in BM on the physical and mental development of infants.

### 4.4. Metabolic Hormones: Leptin, Adiponectin, Ghrelin, Insulin

Leptin is an adipocyte-derived hormone, which functions by reducing appetite and increasing energy expenditure. Its levels correlate with fat mass in adults and children [84]. It is also present in BM; it is produced and secreted by mammary epithelial cells in milk fat globules and transferred from the blood to milk by secretory epithelial cells [85,86]. Its concentration is higher in colostrum than in transitional milk and only moderately correlated with maternal and infant weight or body mass index [87]. Studies show that BM leptin could not only impact the short-term control of food intake in neonates as a satiety signal but also affect energy balance and body weight regulation in the long term [88]. Studies on animals suggest that the oral administration of leptin at doses found in milk can reduce food intake. Moreover, it promotes the formation of neural circuits controlling food intake and adiposity further in life and may also regulate body weight gain during the first months of an infant’s life. Rats given physiological amounts of leptin in BM showed lower body weight and fat content [89]. However, the exact roles of BM leptin in infant development are still unclear and need further studying as many studies analyzing its effects on neonates present contrasting data [90].

Adiponectin is an adipose-specific protein regulating lipid and glucose metabolism as well as inflammatory response and is responsible for the stimulation of food intake and reduction in energy expenditure. Its levels are inversely related to adiposity and positively associated with insulin sensitivity. They are lower in individuals with obesity and type 2 diabetes [91,92]. Adiponectin is synthesized mainly in the adipose tissue, but some studies show that it can also be synthesized by the mammary epithelial cells [93]. Its concentration in BM is higher than leptin’s and it decreases with the duration of breastfeeding [89]. The more-active adiponectin dominates in BM, and it may lead to the enhancement of insulin sensitivity and suppression of inflammation. It was observed that high levels of adiponectin in BM have an inverse relationship with the adiposity of infants [93].

Ghrelin, called the “hunger hormone”, is an amino acid produced in the secretory cells of the stomach. It is involved in energy balance regulation and stimulates food intake and appetite, increases gastric acid secretion, and improves gastrointestinal motility [89,90]. Some studies show that ghrelin levels in BM are significantly higher than in the serum [90]. It was also reported that active ghrelin concentration correlates positively with infant growth rate and weight gain [91]. Thus, ghrelin content in BM may be a factor through which breastfeeding could influence infant feeding behavior and body composition [89].

Insulin is produced by β-cells located in the pancreatic tissue. It is an anabolic hormone and promotes the cellular intake of glucose in muscles and in adipose tissue [18]. Insulin is present in BM. Its primary source is maternal blood but there is evidence that mammary epithelial cells can also produce insulin. Its concentration is highest in the colostrum and then decreases to levels detected in maternal serum [92]. Studies show that the intermediate levels of insulin in BM are associated with lower infant weight and lean mass. However, higher levels might be negatively associated with body weight [94,95].

The concentration of metabolic hormones in BM is possibly an important factor influencing infants’ body weight, fat content, and feeding behavior. However, previous studies have shown contrasting data; thus, this matter needs further investigation.

### 4.5. Melatonin

Melatonin is secreted by the pineal gland. Its functions include the adjustment of sleep and circadian rhythm but also antioxidation, anti-inflammatory, anti-apoptosis, and immunomodulatory. It is also present in BM, and it shows fluctuations in its levels depending on the time of the day—its levels are high at night—peaking at around 03:00, and undetectable amounts during the day. It is an important indicator of the synchronization of an infant with the mother’s rhythm as newborns have inadequate amounts of self-produced melatonin—the first detectable circadian rhythm shows around 3 months of age [80,96].

Scientists found that BM melatonin is associated with improved sleep duration and efficiency. The intake of melatonin by newborns helps to keep their physiological mechanisms in a circadian manner, which is important regarding, for example, cardiovascular health. Thus, BM melatonin may have a positive effect on future cardiovascular events. Moreover, melatonin is a powerful antioxidant involved in regulating inflammation. It can also reestablish the balance of the gastrointestinal microbiota and significantly improve metabolic disturbances [97]. Therefore, it is one of the most important ingredients of BM as it plays a role not only in the circadian rhythm of the infant, but it may also affect immunity and the incidence of cardiovascular diseases later in life.

**Table 3 nutrients-16-01487-t003:** Summary of possible effects of hormones included in BM.

Hormone	Possible Effect on BM	Source
Estrogens (E1, E2, 23)	Positive correlation with infant developmental score.	[68,70]
Phytoestrogens	Impaired immune function and the reduction in weight and adipose tissue.	[68,72]
Progesterone	Negative correlation with infant developmental score and the reduction in infant weight.	[70]
Testosterone	Neuro- and immunomodulation and the increase of hemoglobin and hematocrit levels.	[74]
T4, T3	The delay of clinical the recognition of hypothyroidism.	[76]
GCs (cortisol, cortiosne, corticosterone)	Impact on healthy growth, development, infant temperament, fear reactivity, the improvement of the intestinal microbiome and immune system, and lower levels of sIgA.	[78,79,81,82,83]
Leptin	Satiety signal, the impact of energy balance and body weight regulation, and lower body weight.	[88,89]
Adiponectin	The enhancement of insulin sensitivity, suppression of inflammation, and lower adiposity.	[93]
Ghrelin	Positive correlation with infant growth rate and impact on feeding behavior and body composition.	[89,91]
Insulin	Intermediate levels—lower infant weight High levels—higher body mass	[94,95]
Melatonin	Improved sleep duration and efficiency, circadian manner of physiological mechanisms, and the improvement of metabolic disturbances.	[80,96,97]

## 5. Carbohydrates

Lactose is the main carbohydrate present in almost all mammalian milk, which plays a crucial role as the primary energy source in the first year of life. It provides 40% of the energy value to an infant [98]. This disaccharide is the second most prevalent element of breast milk after water and constitutes 7% of the HBM composition (approx. 70 g/L) and its concentration varies among women. Lactose synthesis occurs within the Golgi apparatus of mammary epithelial cells. It is facilitated by the lactose synthase (LS) enzyme complex, which catalyzes the formation of lactose from its precursors, glucose and UDP-galactose. The LS enzyme complex comprises galactosyltransferase, which is intricately associated with α-lactalbumin [99]. There are some theories as to why lactose is a major sugar present only in milk; some propose that it has the solubility necessary in the process of milk synthesis and secretion. It is postulated that lactose grants an equitable amount between glucose and galactose for the host since glucose serves as an energy source, and galactose is a factor contributing to the process of brain development [22]. It is responsible for maintaining osmotic pressure in breast milk because it draws water into intracellular secretory vesicles, and in this way, it regulates HBM volume [100]. Lactose synthesis is the most significant factor regulating HBM production; a higher lactose concentration corresponds with higher 24 h milk volume and more frequent breastfeeding. There are also scarce amounts of monosaccharides (glucose and galactose) found in breast milk [101]. 

Human milk oligosaccharides (HMOs) are pivotal bioactive components in HBM, fourth most abundant after water, lactose, and lipids, and their concentration ranges from 20 to 25 g/L in colostrum and 5 to 20 g/L in mature milk [102]. Every breastfeeding mother produces a distinct and individualized set of oligosaccharides, impacted by maternal genetics, such as Lewis blood group status [100,103]. Compared to other mammalian milk, HMO levels are 100–1000 higher than in the milk of any domesticated farm animal [104]. HMOs consist of five fundamental components, comprising an acid monosaccharide, specifically sialic acid (Sia) or *N*-acetylneuraminic acid, an amino sugar referred to as GlcNAc, and three monosaccharides—L-fucose (Fuc), D-galactose (Gal), and D-glucose (Glc). Currently, more than 200 different HMOs have been identified [105].

HMOs show significant resilience against enzymatic digestion within the gastrointestinal tract. This quality allows them to reach the far end of the gastrointestinal tract in a mostly unchanged state, where resident bacteria can metabolize them [103]. They have been shown to provide additional advantages for an infant. They play a role in regulating neonatal immunity by influencing the responses of both host epithelial and immune cells within the newborn’s gut. Lacto-N-neotetraose (LNnT) is an abundant HMO in human milk. Its level decreases with time and has unique beneficial effects for infants. It has prebiotic characteristics, and its other health effects have also been verified, including anti-inflammatory and immunomodulatory properties, preventing necrotizing enterocolitis, microbial adhesion, antiviral activity, and promoting the maturation of intestinal epithelial cells. Evaluation and clinical trials proved that LNnT, synthesized via chemical, enzymatic, or cell factory methods, is safe for infants and could even be added as a functional ingredient in infant formula [99]. Commensal bacteria in the infant gut digest HMOs, producing short-chain fatty acids (SCFAs), which play a crucial role in establishing a stable gut ecosystem by modulating the immune system, enhancing the gut epithelial barrier function, and driving the development of regulatory T cells, which helps limit intestinal inflammation. Additionally, SCFAs (precisely butyrate) serve as an energy source for colon epithelial cells and contribute to inhibiting potential pathogens. They can also regulate gene expression, inhibit histone deacetylase, modulate interferon-γ production, and suppress nuclear factor κB (NF-κB) activation in human colonic epithelial cells, a mechanism relevant to inflammatory bowel diseases (IBDs) [100]. HMOs can alter an infant’s gut microbiota and are associated with specific bacteria abundance, namely *Bactericides* spp., *Clostridium* spp., and *Lactobacillus* spp., belonging to the physiological gut microbiome. They work as a probiotic since they cause the strong intestinal growth of beneficial Bifidobacteria. They can also inhibit the growth of pathogenic species (*Campylobacter* spp., *Norovirus* spp., *E. coli*, *Entamoeba histolytica*, and *Candida albicans*) [100,105].

## 6. Lipids

Lipids are HBM’s main energy source ingredient, covering almost 50% of an infant’s daily nutritional supply and constituting 3.5–4.5% of breast milk [12]. Between birth and 6 months of age, infants who are exclusively breastfed typically consume an average of 21.42 g per day of the lipids present in human milk [106]. During the early months of lactation, the typical lipid content in human milk stays relatively stable. However, there is noteworthy diversity not only among different individuals but also within the same individual over time, particularly regarding the concentration of milk fat. Among the various macronutrients found in milk, fat displays the most prominent fluctuations in concentration. The level of milk fat tends to increase with prolonged breastfeeding sessions and experiences fluctuations over a day. Furthermore, the amount of fat in HBM rises with an extended time interval between the current milk expression and the previous one from the same breast. Interestingly, in each breastfeeding session, the milk fat content gradually rises, with hindmilk (at the end of feeding) containing significantly higher fat levels compared to foremilk (at the beginning of the feed); this might lead infants with greater appetite to receive milk with increased fat and energy value, compared to less peckish ones receiving milk richer in essential water-soluble substances. Additionally, there is a correlation between milk fat content and maternal diet; fat deposition during pregnancy, weight fluctuation, and an increase in maternal adiposity may lead to increased secretion of lipids in HBM [106,107]. 

The milk fat is found in HBM in the form of small globules, with an average diameter ranging from 3 to 5 μm. The globules consist of lipids, primarily triacylglycerols (95–98%), concentrated in the core, and less numerous amphipathic particles—phospholipids and cholesterol—creating the globule surface [12,108]. There are also small doses of free fatty acids (FAs)—mono- and diacylglycerols [109]. Triglycerides (TAGs), which are hydrophobic lipids, are synthesized in the endoplasmic reticulum using fatty acids from the maternal bloodstream, along with predominantly intermediate-chain fatty acids with 12 and 14 carbon atoms synthesized from acetyl-CoA. After being released from the endoplasmic reticulum of the mammary epithelial cells into the cytosol, the core rich in triglycerides is enveloped by an inner membrane originating from the endoplasmic reticulum. This membrane consists mainly of a monolayer composed of phosphatidylethanolamine, phosphatidylserine, phosphatidylinositol, and cholesterol. As these lipid droplets are further excreted from mammary epithelial cells into the alveolar space, they are surrounded by a segment of the apical plasma membrane. This process adds another phospholipid bilayer, forming a phospholipid trilayer, along with the inclusion of other constituents from the mammary epithelial cell membrane, such as membrane proteins and glycoproteins [107].

Fatty acids found in HBM are known for having many principal effects on infants, including central nervous system development, lipid metabolism, membrane composition and function, immune system, digestion, and infant growth [107]. Human breast milk lipids’ effect on infants is summarized in Figure 2.

They are the precursors of important tissue hormones—eicosanoids—prostacyclins, prostaglandins, thromboxane, and leukotrienes, crucial for proper cell responses [109]. They also carry lipid-soluble vitamins [100].

For infant brain development and function, a dietary supply of polyunsaturated fatty acids (PUFAs) is indispensable. In the first 1000 days of an infant, there is a rapid increase in brain volume. ARA and DHA, the most prevalent long-chain polyunsaturated fatty acids (LCPUFAs) are present in neuronal membranes, since approximately 60% of the dry weight of the human brain comprises lipids, with about 35% of the lipids in the grey matter being LCPUFA. LCPUFAs undergo rapid accumulation in the human brain during the initial 1000 days, alongside a significant increase in brain volume. They also play essential roles in influencing the physical properties of membranes and contribute uniquely to the development and functioning of the brain; these processes encompass the modulation of neural metabolism, differentiation, plasticity, neuroprotection, and anti-inflammatory effects [30,110]. Other PUFAs contributing to infant nervous system development are α-linolenic acid (ALA) and linoleic acid (LA). They play an important role in brain and eye maturation, as well as internal organ development. An important lipid that plays a role in brain development is sphingomyelin, which plays a crucial role in CNS myelination [100].

Dietary fatty acids can influence an infant’s gut and contribute to its microbiome diversity. Some investigations provide evidence supporting the significance of sphingolipids, especially sphingomyelin and gangliosides, in influencing intestinal development, protection against intestinal injuries as well as regulating inflammation. Sphingomyelin undergoes digestion by nucleotide phosphodiesterase pyrophosphatase 7 (NPP7), an enzyme located in the brush border of the intestinal epithelium. NPP7 exhibits phospholipase C activity against platelet-activating factor, a pro-inflammatory lipid mediator produced by gut epithelial cells, associated with inflammatory bowel disease, ischemic colitis, and necrotizing enterocolitis (NEC). Studies have indicated that gangliosides play a regulatory role in moderating pro-inflammatory signals within the intestine and have protective effects on the bowel in an infant model of necrotizing enterocolitis (NEC), which had been verified in a follow-up investigation with rats fed with dietary gangliosides. After inflammation induced by lipopolysaccharide, they showed a diminished expression of pro-inflammatory mediators (prostaglandin E2, LTB4, IL-1β, and TNF-α) in the intestinal mucosa [34].

**Figure 2 nutrients-16-01487-f002:**
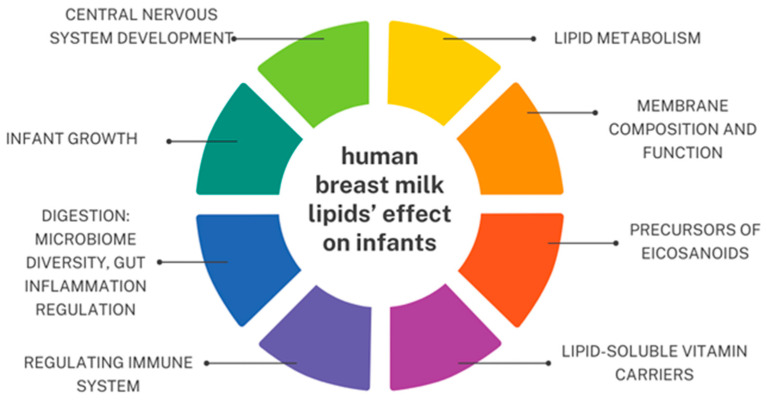
Human breast milk lipids’ effect on infants based on [34,79,109,110]. An original author’s artwork.

## 7. Minerals (Micronutrients)

Human breast milk contains more than 20 minerals, such as iron, copper, and zinc, which are most abundant in colostrum and decrease through the course of lactation. Unlike vitamins, mineral levels in breast milk are generally not affected by maternal status or supplements and remain relatively stable. A summary of the minerals in BM is shown in Table 4.

Worthwhile to emphasize is that breast milk from mothers feeding premature infants is different from breast milk from mothers feeding full-term infants. Most minerals have comparable levels in both preterm and full-term breast milk. However, copper and zinc levels are higher in the human breast milk of mothers feeding preterm infants and decrease gradually with lactation, whereas calcium is lower in preterm cases and gradually increases with lactation [21]. 

Micronutrients are crucial components of human breast milk as they play a fundamental role in healthy brain development and deficiencies during early development can profoundly impact cognitive outcomes [111].

Recent studies indicate that the critical period for nutrition occurs during late fetal and early postnatal life. There is growing recognition that achieving optimal postnatal nutrition is closely tied to optimizing fetal nutritional status before birth [112,113].

Postnatally in exclusively breastfed infants, breast milk serves as the only source of both macro- and micronutrients [111].

### 7.1. Iron 

Iron in milk is found both in the lipid and in the compound of low-molecular-weight peptides. Some small amounts of iron are also connected to lactoferrin [114]. About 10% of the milk iron is bound to casein [115]. This micronutrient is transported by metal transporter 1 through the basolateral membrane and is exported by ferroportin in the apical membrane which, among other epithelial active transporters, regulates milk iron content and is believed to be responsible for insulating milk iron levels from changes in maternal iron status [115]. Iron plays a significant role in oxygen transportation as well as in metabolic processes, being a crucial component of hemoglobin and enzymes, respectively. 

It is also a nutrient with a particularly large effect on early brain development, in processes like myelination or monoamine neurotransmission [112].

Iron levels in breast milk are highest in colostrum (0.5–1.0 mg/L) and decrease as lactation continues (0.3–0.7 mg/L in mature milk) [21]. On average, iron levels are around 1 mg/L during the first week after birth, then decrease to a range of 0.4 to 0.9 mg/L during the first month, followed by further decreases to 0.2 to 0.4 mg/L from 1 to 3 months, and eventually to 0.1 to 0.3 mg/L from 3 to 6 months [115]. The changing levels of iron in HM is shown in Figure 3.

Milk iron concentration is not rich enough to meet an infant’s needs; therefore, supplementation can be advised after 6 months of age [114]. It is an essential remark, as infants grow and develop rapidly; therefore, they are especially vulnerable to the negative effects of iron deficiency. Insufficient iron intake during early development can lead to irreversible impairments in behavioral outcomes [111]. However, during the early stages of life, newborns meet their iron requirements by using the reserves stored in their liver, primarily acquired during the last trimester of pregnancy [114].

The concentration of iron in breast milk is not contingent on the mother’s diet and remains largely unaffected by her own iron levels [114,115]. 

Only a limited number of maternal factors impact the iron concentrations found in breast milk. Maternal intrinsic factors, such as body storage, the length of gestation, along with environmental or infection variables, oral supplementation in anemic mothers [115], and even the rare cases of abnormal body metal availability, did not affect metal transfer from blood serum to breast milk [114].

### 7.2. Zinc

Zinc is involved in numerous cellular functions and is essential for growth and development [116]. Similar to iron and copper, zinc is present in both the whey and fat fractions of human milk [117].

There were notable differences observed between the different stages of lactation. Zinc concentration initially starts at a high level and then decreases rapidly until it reaches a plateau [114,118]. The reported average zinc levels in colostrum range from 4 to 9 mg/L. These levels decrease to 2 to 4 mg/L by 1 to 2 weeks postpartum and stabilize at 1 to 2 mg/L beyond two weeks. Eventually, zinc levels fall below 1 mg/L by 9 months. The levels of zinc are similar between fore- and hindmilk [115]. The mean zinc concentration of breast milk obtained in a total of 242 studies with 37,614 participants was 2.57 mg/L [118]. The estimated mean daily transfer of zinc to the infant via breast milk is approximately 4 mg in colostrum, 1.75 mg at 1 month, and 0.7 mg at 6 months [114]. The changing levels of zinc in HM is shown in Figure 3.

Mothers with inadequate zinc intake may have slightly lower levels of zinc in their breast milk compared to mothers with sufficient intake. However, these differences are not considered clinically significant [115].

Zinc concentrations tend to be higher in adult mothers, the mothers of preterm infants, mothers who exclusively breastfeed, mothers with good nutrition, and mothers living in upper-middle and high-income countries, although only limited data are available [115,118]. No significant relationship has been discovered between breast milk zinc concentrations and maternal factors such as smoking, iron supplementation, multivitamin/mineral supplementation (including zinc), or the length of gestation [114].

**Figure 3 nutrients-16-01487-f003:**
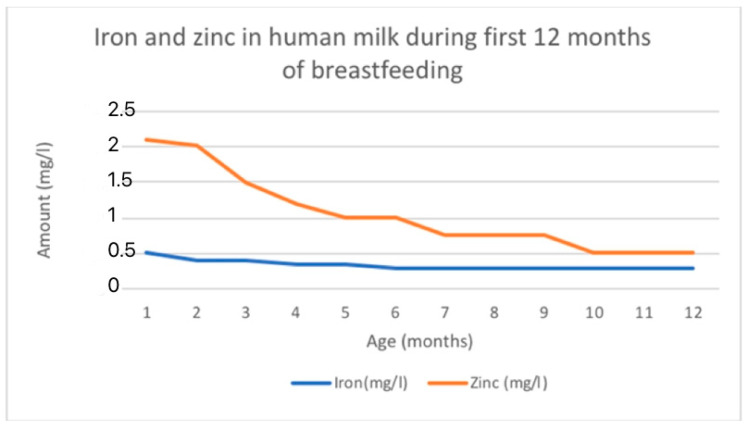
Fe and Zn changing content in human milk according to data adapted from [116].

### 7.3. Calcium

Calcium is essential for bone health and serves a crucial role as a messenger in cell-signaling pathways [114].

The concentration of calcium in human milk can be influenced by the maternal dietary intake of this mineral, especially in geographic areas where the habitual intake of calcium is low [114]. Exclusively breastfeeding infants may not always receive sufficient calcium intake to meet the recommended daily intake [106,119].

Calcium concentration decreases linearly over the duration of lactation [118] and is tightly linked to casein and citrate in the milk [114]. Ca changing content in human milk is shown in Figure 4.

The mean calcium concentration in breast milk obtained from 154 studies with 22,307 participants using inductively coupled plasma mass spectrometry was 262 mg/L [118]. Breast milk total calcium concentrations experience a rapid increase in the first 5 days after birth, followed by a gradual decline for the remainder of lactation. However, ionized calcium concentrations in breast milk remain stable throughout lactation, indicating a homeostasis similar to that observed in blood [114].

Lower calcium breast milk concentration is observed in lactating adolescents and in anemic mothers. There is no such influence indicated for other factors such as the length of gestation, fore- and hindmilk, maternal age, parity, race, lactation history, smoking, and oral contraceptives [114].

**Figure 4 nutrients-16-01487-f004:**
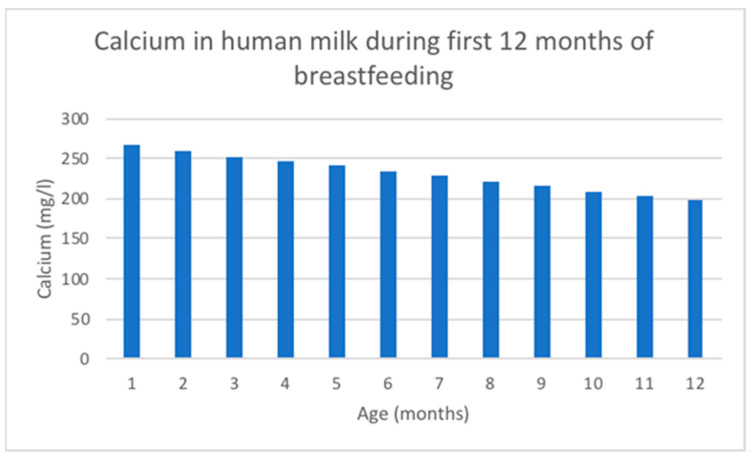
Ca changing content in human milk according to data adapted from [116].

### 7.4. Copper

Copper is a micronutrient involved in cellular respiration as a cofactor for enzymes, and plays an important role in iron metabolism and connective tissue synthesis. Copper accumulates in the fetal liver during gestation and is then released into the circulation during the early stages of neonatal life, so there is no clinical or scientific evidence supporting the need for extra copper, at least during the initial 6 months of life.

The level of copper decreases with the progress of lactation. In breast milk, 20–25% of the copper is carried by ceruloplasmin [114]. Copper also affects the process of myelination in early brain development [111].

The levels of copper, much like iron, are not linked to the mother’s health status, diet, or supplementation. Drugs, age, oral contraceptive use, or supplementation did not affect breast milk copper concentration [114].

### 7.5. Phosphorus

Phosphorus plays a crucial role in various biological processes. It serves as a structural component of cell membranes and nucleic acids. Additionally, phosphorus is involved in important functions such as bone mineralization, cell signaling, energy production, and maintaining acid–base balance within the body [114].

The concentration of phosphorus in breast milk is correlated with the concentration of calcium [119] and does not seem to be affected by maternal intake, age, parity, race, lactation history, sampling techniques, smoking, or oral contraceptive use. Similar to calcium, phosphorus concentrations are highest in early transitional milk and decline gradually as lactation progresses.

Comparing to other mammals, human milk phosphorus concentration is relatively low, which may serve as a mechanism to inhibit the growth of fecal pathogens or protect the immature newborn renal system from disturbances in calcium metabolism [114].

### 7.6. Selenium

Selenium is a vital micronutrient that is incorporated into specific proteins called selenoproteins, which serve various important roles like antioxidant protection, supporting thyroid function, and aiding the immune system [120].

In human milk, selenium is mainly present as a component of an antioxidant enzyme, glutathione peroxidase. It is also found in substances such as selenocystamine, selenocystine, and selenomethionine. Only a minor fraction is associated with the milk fat [117].

Newborn infants have selenium reserves at birth, but they also rely on the selenium provided by their mothers’ milk.

Selenium concentrations are typically high in colostrum; however, they tend to decrease as the lactation progresses. This can be explained by a general decline in the levels of milk proteins into which selenium is incorporated [114].

### 7.7. Magnesium

Magnesium serves a structural function in bone and participates in over 300 vital metabolic reactions within the body [114].

The maternal storage of magnesium is mobilized during lactation and supplies the mammary gland.

The mean magnesium concentration value in human milk ranges from 20 to 40 mg/L. The secretion of magnesium into breast milk does not appear to be influenced by variables, such as adolescent motherhood, gestation length, maternal undernutrition, metabolic disorders, race, socio-cultural diversity, smoking habits, dietary calcium and magnesium intake, or oral hormonal contraceptives. Several researchers, however, have noted slight fluctuations during the initial six months, either increasing or decreasing [114,117].

Since the majority of magnesium in breast milk is bound to low-molecular-weight proteins, there is a minimal disparity in its concentrations between foremilk and hindmilk.

### 7.8. Iodine

Iodine is essential for normal growth, mental development, and the survival of infants [114].

Neonatal iodine deficiency has been associated with compromised mental ability and, in severe cases, cretinism. This might be related to the significant role that iodine plays in the myelination process [111].

Infant requirements are estimated to be 15 mcg/kg daily in full-term infants and 30 mcg/kg daily in preterm infants. A milk iodine concentration, which correlates well with maternal urinary iodine concentration, ranging from 100 to 200 micrograms per liter (mcg/L) has been proposed as an indicator of sufficient iodine status in lactating women [115]. More than 75% of the iodine content in human milk exists in the form of ionic iodide [117].

Systematic reviews and studies on iodine nutrition have indicated that iodine levels in breast milk are generally adequate in countries where iodine sufficiency is maintained naturally. However, in countries where foods are fortified with iodine, many mothers do not acquire sufficient iodine through diet alone, highlighting the need for additional supplementation for lactating mothers [114,115].

Iodine concentration levels reach maximum values in colostrum and gradually decrease as lactation progresses [114]. The quantity of iodine in breast milk can differ depending on the mother’s iodine intake and may also be influenced by genetic factors [114,115,117]. There is no significant difference in iodine values between fore- and hindmilk. Smoking has been observed to have an inverse association with breast milk iodine levels [114]. Mothers who are or will be breastfeeding should avoid or minimize to an extent unnecessary exposure to iodine, as this can increase breast milk iodine levels and cause transient hypothyroidism.

The WHO recommendations state that lactating women have urinary iodine levels of 100 mcg/L [115].

### 7.9. Role of Micronutrients in Child’s Neurodevelopment

Given the significant micronutrient requirements of infants during crucial neurodevelopment stages and the growing evidence suggesting the lifelong impact of nutrition in the first 1000 days of life, this review highlights the importance of conducting future research in this area to improve population health outcomes worldwide [111].

It is widely acknowledged that carotenoids, vitamin B6, and selenium play significant roles in infant neurodevelopment, but they are just a few of the many micronutrients and bioactive components found in breast milk that affect brain formation and function. The impact of numerous other micronutrients in breast milk on infant development is yet to be thoroughly studied [111,112,114,115].

**Table 4 nutrients-16-01487-t004:** Summary of minerals’ impact on child’s development and recommended supplementation. ^1^ BM—breast milk, ^2^ CNS—central nervous system, ^3^ RDA—recommended dietary allowance.

	Iron	Zinc	Calcium	Copper	Phosphorus	Selenium	Magnesium	Iodine
Function	Metabolic processes, oxygen transportation, and myelination in CNS ^2^	Cellular functions; immunity, growth, and development	Bone health and cell signaling	Cellular respiration and iron metabolism; myelination in CNS ^2^	Bone mineralization and cell signaling, and forms cell membranes and nucleic acids	Antioxidant protection, thyroid function, immunity, and cholesterol metabolism	Bone structure and metabolic reactions	Growth, nervous system development, and thyroid hormone synthesis
Level in BM ^1^	Highest in colostrum and decreases over lactation	Stages as follows: high in colostrum; then a rapid decline; plateau phase and further decline	Decreases over lactation (total calcium concentration; ionized form is stable)	Decreases over lactation	High in transitional milk and declines over lactation	High in colostrum and decreases over lactation	Varies between foremilk and hindmilk	Maximum levels in colostrum and decreases over lactation; no difference in values between fore- and hindmilk
The influence of maternal factors on the levels	No influence	No significant relationship	Maternal intake, geographic area, and anemia	No influence	No influence	Strongly influenced by mother’s diet	Not observed	Maternal intake and genetic factors
Comment	Both in the lipid form and small peptides; newborns have their own reserves	Both whey and fat fraction forms in BM; similar values between fore- and hindmilk; high bioavailability	-	Newborns have their own reserves	Relatively low concentration in BM protects the immature renal system of a newborn	The component of glutathione peroxidase in BM; newborns have their own reserves	A majority is bound to proteins in BM	A majority exists in the form of ionic iodide in BM
Recommended supplementation or extra intake for lactating mothers	Usually recommended, but not essential	Increase zinc intake by 50%	A total of 5 servings a day of calcium-rich foods	An amount of +400 μg to RDA ^3^	Not needed	An amount of +15 μg to RDA ^3^	Not needed	A total of 200 μg of potassium iodide a day
Source	[112,114,115,121]	[114,115,116,117,121]	[106,114,118,119,121]	[111,114,121]	[114,119,121]	[114,120]	[114,117]	[114,115,117]

## 8. Vitamins

Human milk contains both water-soluble and fat-soluble vitamins, providing sufficient amounts for typical infant development, except for vitamins D and K. Infants nursed by mothers adhering to a strict vegetarian diet may additionally need vitamin B12 supplementation to prevent deficiencies. Moreover, a single supplementary dose of vitamin A is advised in deficient populations as soon as possible after childbirth. A summary of the effects of vitamins on infant development and the importance of supplementation is summarized in Table 5.

### 8.1. Riboflavin

Free riboflavin and its coenzymatic form FAD are the primary forms present in milk. Other flavins found in milk include 10-hydroxy-ethylflavin, 10-formyl-methylflavin, 7α-hydroxy-riboflavin, 8α-hydroxy-riboflavin, and FMN [117]. The concentration of riboflavin in milk depends on the mother’s diet and supplementation, averaging 0.35 mg/L in healthy women [114].

### 8.2. Vitamin B6

The role of vitamin B6 encompasses participation in metabolic processes as it acts as a cofactor for >100 enzymes, the synthesis of neurotransmitters, the regulation of the immune system, and the production of hemoglobin [114]. Pyridoxal is the primary form of vitamin B6 in milk, with additional forms including pyridoxamine-5′-phosphate, pyridoxine, and pyridoxamine [117]. The concentration of vitamin B6 in milk increases 2–5 times in the first weeks after childbirth. From 2 to 6 months, its levels stabilize, only to gradually decrease later on. The average concentration of vitamin B6 in mature milk (from 2 to 6 months postpartum) in healthy mothers is 800–1200 nmol/L. Both the mother’s diet and supplementation impact the concentration [114,115]. Moreover, a small, cross-sectional study recently revealed a positive correlation between the levels of pyridoxal found in transition milk and infant performance on two subscales (habituation and autonomic stability) of the Neonatal Behavioral Assessment Scale (NBAS) at 8–11 days postpartum (*n* = 25) [122].

### 8.3. Vitamin B12

The role of vitamin B12 is multifaceted, primarily involving DNA synthesis, neurological function maintenance, and metabolism regulation [114]. Methylcobalamin is the predominant form of vitamin B12 in milk, alongside smaller quantities of deoxyadenosylcobalamin, hydroxocobalamin, and cyanocobalamin [117]. All these variations are tightly bound to apo-haptocorrin, which could potentially interfere with research findings on this vitamin. The average concentration of vitamin B12 in healthy mothers during the first 28 weeks postpartum ranges from 200 to 700 pmol/L. It is highest in colostrum and gradually decreases during the initial 3–4 months of lactation. The concentration is closely linked to the mother’s diet. Maternal vegetarianism and pernicious anemia negatively impact the vitamin’s concentration in milk. There was some indication of a positive link between the habitual intake of vitamin B-12 by mothers and the concentration of vitamin B-12 in the milk, especially among marginally nourished women [114,115,117,123].

### 8.4. Vitamin C

Antioxidant vitamins present in human milk are vital for immune system regulation. Vitamin C boosts white blood cell activity and antibody production, and promotes the synthesis of interferons [114]. Vitamin C in human milk mainly exists in the form of ascorbic acid (AA) and dehydroascorbic acid (DHAA)—these two forms are biologically significant [117]. The average concentration of vitamin C in mature milk from healthy mothers ranges from 50 to 90 mg/L [115]. The highest concentration is observed in colostrum, maintained for a certain period, and then gradually begins to decrease around 12 months after childbirth. According to data, the routine supplementation of vitamin C does not affect its concentration in women with high socioeconomic status, unlike women with lower status [114].

### 8.5. Vitamin A

Vitamin A is essential for maintaining optimal vision, supporting immune function, and facilitating proper growth and development. The main form of vitamin A in breast milk is retinyl esters, which undergo de-esterification in the child’s digestive system. Vitamin A also occurs in the form of retinol and beta-carotene [117]. It originates from two sources—circulating plasma retinol bound to plasma retinol-binding protein (pRBP) and transthyretin, and from dietary intake (transferred directly to the mammary gland from chylomicrons). The average concentration in healthy mothers is highest in colostrum (800 to 1400 mcg/L), gradually decreasing (to an average of 300 to 800 mcg/L in mature milk). Especially in mature milk, retinol levels are positively correlated with milk fat [114,115,124]. In developed regions, a balanced diet generally offers a sufficient vitamin A intake, whereas in developing countries, it is advised for mothers to receive a single supplementary dose of vitamin A [121].

### 8.6. Vitamin D

Vitamin D is crucial for bone mineralization and growth. It supports the development of a healthy immune system and brain development [114]. Vitamin D in human milk consists mainly of vitamin D2 (ergocalciferol) and vitamin D3 (cholecalciferol), with contributions from their 25-hydroxy metabolites and possibly 24,25-dihydroxyvitamin D and 1,15-dihydroxyvitamin D [117]. Vitamin D concentrations in human milk are low, typically less than 40 IU/L, especially when the mother has limited exposure to sunlight. Diet may affect total vitamin D3, but not active 25(OH)D concentration. It is recommended for lactating mothers and infants to take vitamin D supplements. A correlation between maternal obesity and lower levels of vitamin D has been observed [12,67,115,118].

### 8.7. Vitamin E

Vitamin E acts as an antioxidant and supports immune function [114]. Vitamin E refers to the eight chemically related α-, β-, γ-, and δ-tocopherols and α-, β-, γ-, and δ-tocotrienols [117]. Its most active compound in human milk is α-tocopherol. The concentrations of vitamin E are highest in colostrum (average alpha-tocopherol levels range from 20 to 50 micromoles/L), decrease as the milk matures, and stabilize after the first month of lactation (levels in mature milk range from 3 to 9 micromoles/L). Preterm birth, maternal obesity, and smoking are associated with lower milk vitamin E levels. The correlation between dietary intake and vitamin E concentration in breast milk has not been proven. Maternal vitamin E supplementation can increase vitamin E levels [114,117].

### 8.8. Vitamin K

Vitamin K is pivotal for the activation of various coagulation factors essential for hemostasis, thereby preventing hemorrhagic complications. Vitamin K in human milk is mostly presented as phylloquinone (vitamin K-1) and menaquinone-4 (vitamin K-2). Menaquinone-6 has been found in trace amounts [117]. Vitamin K concentration in human milk is very low (1 to 9 mcg/L) due to poor placenta traversal. Therefore, vitamin K supplementation is recommended after birth. The breast milk concentrations of vitamin K are not associated with maternal dietary intake. They are, however, affected by supplementation [12,114,115].

**Table 5 nutrients-16-01487-t005:** A summary of the effects of vitamins on infant development and bioactive forms. *—retinol activity equivalents (RAEs).

Vitamin	Main Forms	Function	Recommended Dietary Allowance for Infants (0–6 msc.)	Risk of Inadequacy	Sources
Riboflavin	free riboflavin and conzymatic form FAD	energy metabolism, and part of co-enzymes FMN and FAD in redox reactions	0.3 mg	-	[114,117,125]
Vitamin B6	pyridoxal	neurological development, the synthesis of neurotransmitters, immune system regulation, and hemoglobin production	0.1 mg	-	[111,114,117,125]
Vitamin B12	methylcobalamin	cofactor in DNA synthesis and folate metabolism and crucial for maintaining neurological function and metabolism regulation	0.4 mcg	the infants of vegan mothers are at risk of deficiency and might need supplementation	[114,115,117,123,125]
Vitamin C	ascorbic acid and dehydroascorbic acid	antioxidant, boosts white blood cell activity and antibody production, and promotes the synthesis of interferons	40 mg	-	[114,115,117,126]
Vitamin A	retinyl esters	optimal vision, immune function, growth and psychomotor development, and the prevention of obesity and type 2 diabetes mellitus	400 mcg RAE *	in developing countries deficient in vitamin A, mothers require supplementation	[114,115,117,121,124,127]
Vitamin D	ergocalciferol and cholecalciferol	proper bone mineralization and growth development, and blood pressure and glycemia regulation	10 mcg; 400 IU	the concentrations in breast milk are insufficient to meet the daily dietary needs of an exclusively breastfed infant. Supplementation is necessary	[12,67,114,115,117,118,128,129,130]
Vitamin E	α-tocopherol	antioxidant and immune system regulation	4 mg	-	[114,117,126]
Vitamin K	phylloquinone and menaquinone-4	cofactor for the synthesis of coagulation factors II, VII, IX, X, and proteins C and S	2.0 mcg	poor placenta traversal elevates the risk of vitamin K deficiency bleeding (VKDB). To prevent VKDB, a single, intramuscular dose of vitamin K1 at birth is recommended	[114,115,117,131,132]

## 9. Conclusions

Human milk is an amazing example of bioactivity and the perfect adaptation of the mother’s body to her baby’s needs. Understanding the processes of human milk secretion and production, as well as the factors influencing changes in its composition and concentration, seems to be of inestimable value to modern medicine. As emphasized repeatedly in this article, milk represents the only optimal food for the newborn, adapted to its needs, actively changing, and diversified. It is not only a source of the supply of essential substances for the development of the young child but also has an immunological and immune-stimulating function, particularly important in the case of newborns.

The most immunologically important factor of milk is sIgA antibodies, which are crucial in the first period of a newborn’s life. They mainly protect the mucous membranes of the respiratory and gastrointestinal systems, which are particularly vulnerable to infection in the face of an immune system that is just forming, stimulated to develop by the interleukins contained in milk.

The HMOs present in breast milk regulate neonatal gut immunity by promoting the maturation of intestinal epithelial cells and preventing inflammations (such as IBDs and necrotizing enterocolitis); they can alter an infant’s gut microbiota by working as a prebiotic, and inhibit the growth of pathogenic species. SCFAs, the products of HMO bacterial digestion, regulate intestinal gene expression, drive the development of regulatory T cells, and serve as an energy source for epithelial cells.

Lipids play a key role in the neurological advancement of infants. Sphingomyelin is responsible for the myelination processes within the central nervous system. LCPUFAs are the constituents of neuronal membranes, with major effects on the development of gray matter. They contribute to cerebral metabolism, cellular differentiation, synaptic plasticity, neuroprotection, and the mitigation of inflammatory responses. LCPUFAs significantly influence ocular maturation.

Dietary fatty acids can impact the composition of an infant’s gut bacteria, promoting a diverse microbiome. Sphingolipids are important in shaping intestinal growth, shielding against damage to the intestines, and regulating inflammation. Sphingomyelin and gangliosides help regulate inflammatory signals in the gut and provide protective effects by reducing the expression of pro-inflammatory mediators in the intestinal mucosa.

Micronutrients play a significant role in brain development, essential metabolic processes, cell signaling, growth, and bone health. Additionally, micronutrients support the immune system and provide antioxidant protection.

Vitamins play a pivotal role in the healthy growth and development of infants, exhibiting a broad spectrum of functions essential for their overall well-being. For instance, vitamin A is crucial for promoting healthy vision and supporting immune function, while vitamin D aids in the absorption of calcium and promotes proper bone development. Additionally, vitamin C serves as an antioxidant, assisting in the formation of collagen, which is vital for the growth and repair of tissues. Moreover, B vitamins are essential for proper brain development and function.

Hormones contained in BM affect many systems of infants. They affect the nervous system, endocrine system, immune system, the development of intestinal microbiome, and gastrointestinal system, having an influence on, among others, infant weight, feeding behavior, or sleep duration. All these aspects are extremely important when it comes to the child’s further development and health; thus, BM hormones’ role and possible use is a promising field for studying.

The influence of growth factors on the newborn body systems has already been described in the paragraph dedicated to them. Mostly, even if some factors are seemingly dedicated to specific systems, they usually act strongly synergistically with each other on various systems.

It is worth noting that the systems on which factors act are strongly dictated by the way the substance enters the body. Factors produced by the infant or transported from the mother during fetal life act more holistically. Orally supplemented factors or these taken with mother’s milk act more locally. In this context, they act mostly in the digestive system. They support its proper development and support it in pathological situations, e.g., in the case of premature babies who have not yet developed.

As highlighted in the work, currently there are no ideal mixtures that can replace human milk. However, detailed, advanced diagnostics and research on this topic could help scientists create compositions that will imitate human milk sufficiently, at least to a small extent. Undoubtedly, over time, more and more well-balanced and refined mixtures will appear, imitating the ideal composition of micro and macro elements that is human milk. Finally, it is worth adding that promoting breastfeeding following the WHO recommendations should be one of the priorities of pediatric and neonatal care. Breastfeeding should be treated not only as a form of nourishing the infant but also as an essential preventive measure in physical and mental development and the prevention of many serious diseases.

## Figures and Tables

**Table 1 nutrients-16-01487-t001:** Selected cytokines and their effects on human milk bioactivity.

Cytokine	Possible Effect on BM	Source
IL-2	regulates the growth and differentiation of the T lymphocytes and NK cells which are transported to the BM	[33]
IL-6	has both pro-inflammatory and anti-inflammatory properties, stimulates mammary gland epithelial cells to increase sIgA antibody transport, and stimulates the neonatal body’s own antibody production by inducing follicular T helper cells in Peyer’s patches	[14,33,36]
IL-8	leukocytes recruitment in the maternal organism and flow into the milk by chemotactic properties	[33,37]
IL-10	stimulates the expansion of B lymphocytes and the regulation of the immune response in the infant gut area, and promotes the formation of a physiological microbiota	[13,34,36]
TGF-β1 TGF-β2	immunoregulation by inhibiting naïve T cells from differentiation into Th1 and Th2 and maintaining the differentiation of Treg	[13,33,38]

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
