# Peer review of "Bioactive Components of Human Milk and Their Impact on Child’s Health and Development, Literature Review"

_nutrients, 2024, doi:10.3390/nu16101487_

Round 1
Reviewer 1 Report
Comments and Suggestions for Authors
Line 14 – Abbreviations should be avoided in the Abstract. This section needs to be rewritten once it is not explained how the search was conducted, the type of review and the main results and conclusions. You can’t conclude the abstract mentioning “The study is based on more than 100 articles se- 20 lected from the literature on the subject.” – You have to provide the conclusions and directions for future investigations.
Lines 25-26 – “Human milk is the only food source that can provide a newborn with all the necessary substances for proper development in the first months of life.” – References are needed.
The figures need to be mentioned in the text.
Lines 46-49 – Why Poland? You should provide a worldwide perspective.
References and copyright have to be provided for figures 1 and 3.
The study is interesting and provides relevant information. However, I suggest creating tables mentioning the various studies analyzed and discussed to facilitate reading.
I also find the manuscript to be very descriptive. It would be interesting for the authors to identify practical studies to discuss within each section.
At the end, the numbering of references is in duplicate. Please, correct it.
Comments on the Quality of English LanguageOnly minor edition is recommended.
Author Response
Dear Reviewer,
Thank you very much for taking the time to review this manuscript. Please find the detailed responses in the attached PDF file and corresponding corrections highlighted in re-submitted files.

Reviewer 2 Report
Comments and Suggestions for Authors
The study presented the current state of knowledge about the bioactive components of human milk and their impact on the growth, development and health of the young organism. The study is detailed and accurate. Specific recommendations are as follows.
1.The study analyzed the possible mechanism of bioactive components of human milk in detail. It is recommended to include more clinical studies to enhance clinical significance.
2.The study brought up a very interesting question: what components should be supplemented to complement the child's diet and immunological role of human breast milk. A brief answer to this question is suggested to add to the review.
3.Different bioactive components of human milk have certain effects on the development and protection of the nervous system, digestive system, and immune system of children. It is recommended to briefly summarize these effects according to the system, in order to highlight the benefits of breast milk from different perspectives.
4. Figure 2 is too concise and can provide limited information. It is suggested to enrich the content, such as appropriately increasing the role of each factor in the figure.
Author Response

(The authors gave the same response as above.)

Round 2
Reviewer 1 Report
Comments and Suggestions for Authors
The abstract is still unsatisfactory. The authors should follow the recommendations I provided previously.
Comments on the Quality of English LanguageMinor editing of English language required.
Author Response
Esteemed Reviewer,
As recommended, we have considerably rebuilt the abstract with reference to the recommendations in the review. We hope it is now properly written.